# Elevated ANG2/ACE2 and immune responses associated with *Plasmodium falciparum* and SARS-CoV-2 coinfection in Cameroon

Eric Berenger Tchoupe[1,2], Mary Ngongang Kameni[1,3] MacDonald Bin Eric[1,4], Jean Bosco Taya[1,3], Severin Donald Kamdem[1], Leonard Numfor Nkah[1], Vicky Ama Moor[2], Arnaud Tepa[2], Fuh Roger Neba[1], Anthony Afum-Adjei Awuah[5,6], John Humphrey Amuasi[5,6], Palmer Masumbe Netongo[1,4,7]*

**1** Molecular Diagnostics Research Group (MDRG), Biotechnology Centre-University of Yaoundé I, Yaounde, Cameroon, **2** Department of Medical Biochemistry, Faculty of Medicine and Biomedical Sciences, University of Yaounde I, Yaounde, Cameroon, **3** Department of Microbiology, Faculty of Sciences, University of Yaounde I, Yaounde, Cameroon, **4** Department of Biochemistry, Faculty of Sciences, University of Yaounde I, Yaounde, Cameroon, **5** Kumasi Centre for Collaborative Research in Tropical Medicine, Kwame Nkrumah University of Science and Technology, Kumasi, Ghana, **6** Department of Infectious Diseases Epidemiology, Bernhard Nocht Institute for Tropical Medicine, Hamburg, Germany **7** Biology Program, School of Science, Navajo Technical University, Crownpoint, New Mexico, United States of America

\* masumben@gmail.com

## Abstract

Malaria and COVID-19 co-infections pose a major clinical challenge, as overlapping symptoms can lead to misdiagnosis and delays in treatment. Emerging evidence suggests that SARS-CoV-2 may influence malaria pathogenesis through dysregulation of the renin–angiotensin system. This study assessed clinical, biochemical, and immunological alterations associated with single and dual infections. A total of 96 participants aged 15–64 years were enrolled and classified into four groups: COVID-19 (n = 28), malaria (n = 28), co-infection (n = 16), and healthy controls (n = 24). Blood and nasopharyngeal samples were tested using rapid diagnostic tests, microscopy, and RT-PCR. Disease severity biomarkers were quantified using spectrophotometry and ELISA. Statistical analyses were performed in GraphPad Prism version 9.0, with significance set at p < 0.05. Co-infected participants exhibited significantly elevated biochemical markers (ALT, AST, urea, creatinine, and erythropoietin) compared to all other groups. Co-infection also triggered robust increases in IFN-γ and IL-1β, whereas malaria alone was associated with higher IL-6, IL-4, and IL-10, and COVID-19 alone was associated with elevated IL-2 and TNF-α. ANG2 levels were highest in both COVID-19 and co-infected groups, while ACE2 was markedly elevated in COVID-19 (p < 0.01). Correlation analyses revealed distinct biomarker networks driven by parasitaemia and viral load, implicating pathways linked to inflammation, erythropoiesis, and endothelial dysfunction. Notably, ACE2 demonstrated strong discriminatory power for predicting disease severity, with AUCs of 0.77

**Data availability statement:** All relevant data are within the paper and its Supporting Information files.

**Funding:** Through John H. Amuasi and Palmer M. Netongo, this work was sponsored by the African coaLition for Epidemic Research, Response and Training (ALERRT) which is part of the EDCTP2 Programme supported by the European Union under grant agreement RIA2016E-1612. ALERRT is also supported by the United Kingdom National Institute for Health Research and the Wellcome Trust (Ref 221012/Z/20/Z). The funders had no role in the design, data extraction and analysis, decision to publish, or preparation of the manuscript.

**Competing interests:** The authors have declared that no competing interests exist.

for malaria and 0.85 for COVID-19. These findings underscore the diagnostic and prognostic value of vascular and immune biomarkers for early risk stratification, particularly in malaria–COVID-19 co-infection, and may guide improved clinical management in co-endemic regions.

## Introduction

Malaria is a tropical disease caused by parasites of the genus *Plasmodium* transmitted to humans through the bites of infected *Anopheles* mosquitoes, and according to the WHO's 2024 World Malaria Report the global burden in 2023 reached approximately 263 million cases and 597,000 deaths [1]. COVID-19, caused by SARS-CoV-2, has profoundly affected global health since its emergence in December 2019, spreading to more than 219 countries and resulting in over five million deaths [2]. SARS-CoV-2 is an enveloped virus with a~29.9-kb positive-sense RNA genome that shares about 80% sequence similarity with SARS-CoV [3]. In Cameroon, malaria in 2023 caused an estimated 7.3 million cases and about 11,600 deaths, while by early 2024 the WHO COVID-19 dashboard and national economic data reported roughly 125,000 confirmed COVID-19 cases and approximately 1,970 deaths, reflecting a substantially lower documented burden of COVID-19 compared to malaria [4]. Malaria and COVID-19 are major public health concerns worldwide, and their co-infection is particularly worrisome in tropical regions like Cameroon. COVID-19 and malaria share similar manifestations such as headache, fever, cough, and fatigue [4,5]. This overlap in clinical features can result in misdiagnosis, with patients potentially being incorrectly identified as having one disease instead of the other. SARS-CoV-2 uses ACE2 as an entry receptor via its spike (S) protein and relies on the cellular serine protease TMPRSS2 for activation [6,7]. Furthermore, angiotensin II (ANG II) has been reported to inhibit the development of malaria parasites within mosquito salivary glands by disrupting their membranes. The malaria *Plasmodium* parasite invades red blood cells through interactions involving ACE2 receptors on their surface, with ANG II potentially contributing to this process [8]. Previous exposure to infectious diseases such as malaria may induce long-lasting innate immune protection against endemic pathogens, which could partly explain the lower severity of COVID-19 observed in many African countries despite fragile healthcare systems [9]. Early malaria infections can trigger a surge of pro-inflammatory cytokines, including IL-1, IL-6, and IFN-γ, produced by antigen-presenting cells, which help inhibit parasite growth [10]. Additionally, pro-inflammatory cytokines released by natural killer (NK) cells play a crucial role in antiviral immunity. *In vitro* studies have shown that NK cells can directly combat SARS-CoV-2 infection and reduce tissue fibrosis [11]. Malaria-induced immune activation may provide cross-protection against other infections, thereby potentially reducing the severity of COVID-19 in malaria-endemic regions [12,13]. Both COVID-19 and malaria are systemic infections known to induce multi-organ dysfunction, particularly affecting the liver, kidneys, and haematological system. In malaria, *Plasmodium* infection causes haemolysis, anaemia, thrombocytopenia, and elevated liver

enzymes due to hepatocellular injury and immune-mediated inflammation [14]. Similarly, SARS-CoV-2 infection can trigger hepatic dysfunction through viral replication in hepatocytes and cytokine-induced damage, leading to increased AST, ALT, and bilirubin levels [15]. COVID-19–associated acute kidney injury (AKI) has been linked to direct viral invasion of renal tissue, hypoxia, and cytokine storm effects [16]. Haematological abnormalities, including lymphopenia, elevated D-dimer, and altered platelet counts, are frequent in both infections and often correlate with disease severity. Co-infection may therefore amplify inflammatory and endothelial responses, exacerbating hepatic, renal, and haematological dysfunctions. However, data from Cameroon on these immunopathophysiological interactions remain limited. This study aims to determine how malaria-COVID-19 co-infection influences cytokine, vascular, and biochemical markers compared with mono-infections. The findings will provide new insights into the mechanisms underlying co-infection and generate locally relevant evidence to support improved diagnosis and management of these diseases in malaria-endemic settings.

## Materials and methods

### Ethical considerations

The research complied with national and international regulations, including the CIOMS Guidelines and the Declaration of Helsinki, and received approval from the relevant ethics committees, such as the Centre Regional Delegation of Public Health (Ref. No.: 2020/07/1265/CE/CNERSH/SP). Informed consent was obtained from all participants, with confidentiality assured and measures taken to minimize risks and adverse effects.

### Study design and site

A cross-sectional study was conducted in two phases within the Mfoundi Division of the Centre Region of Cameroon as part of the Clinical Characterization Protocol (CCP) for the COVID-19 Project. Phase 1 was implemented between December 6, 2020, and July 7, 2021, while phase 2 between September 27 and December 18, 2022. The study was carried out in four major health facilities in Yaoundé: the Baptist Hospital, Red Cross Hospital, Central Hospital, and Djoungolo Hospital. Yaoundé, the capital city of Cameroon, is located in the densely populated Centre Region, which hosts over three million inhabitants, with approximately 1.1 to 1.3 million residing in the city itself [17]. The city lies between latitudes 3°45′50″ and 3°59′55″ N and longitudes 11°22′40″ and 11°30′25″ E, covering an area of about 304 km². Locally known as *Ongola* in the Béti language and popularly referred to as the "City of Seven Hills [18]," Yaoundé has a humid equatorial climate characterized by an average temperature of 22°C, relative humidity of approximately 98%, and wind speeds averaging 3 km/h [19]. Malaria transmission in this region is holoendemic and seasonal [20,21], making it a relevant setting for investigating malaria–COVID-19 co-infection dynamics.

### Study population and data collection

Participants were selected and clustered as shown in Fig 1 below. We included participants aged 15 years and above who voluntarily provided written informed consent and tested positive for either malaria or COVID-19 were enrolled in the study. Before inclusion, each participant received an information leaflet explaining the study objectives, potential benefits, and associated risks. From a total of 448 individuals initially recruited, 96 participants were selected for analysis and categorized into four groups: malaria-positive (n = 28), COVID-19–positive (n = 28), malaria/COVID-19 co-infected (n = 16), and healthy controls (n = 24). Pregnant women and individuals with chronic conditions such as HIV infection, diabetes, hepatitis, nephritis, or cancer were excluded to minimize confounding effects on biological markers. Trained clinical personnel collected participant data using a standardized Case Report Form (CRF) which recorded demographic information, medical history, clinical symptoms and with their severity, comorbidities, initial diagnosis, prescribed medications, laboratory results, and the care pathway (hospitalization or outpatient management). This structured approach ensured consistent data collection and enabled a comprehensive assessment of malaria and COVID-19 interactions across the different patient groups.

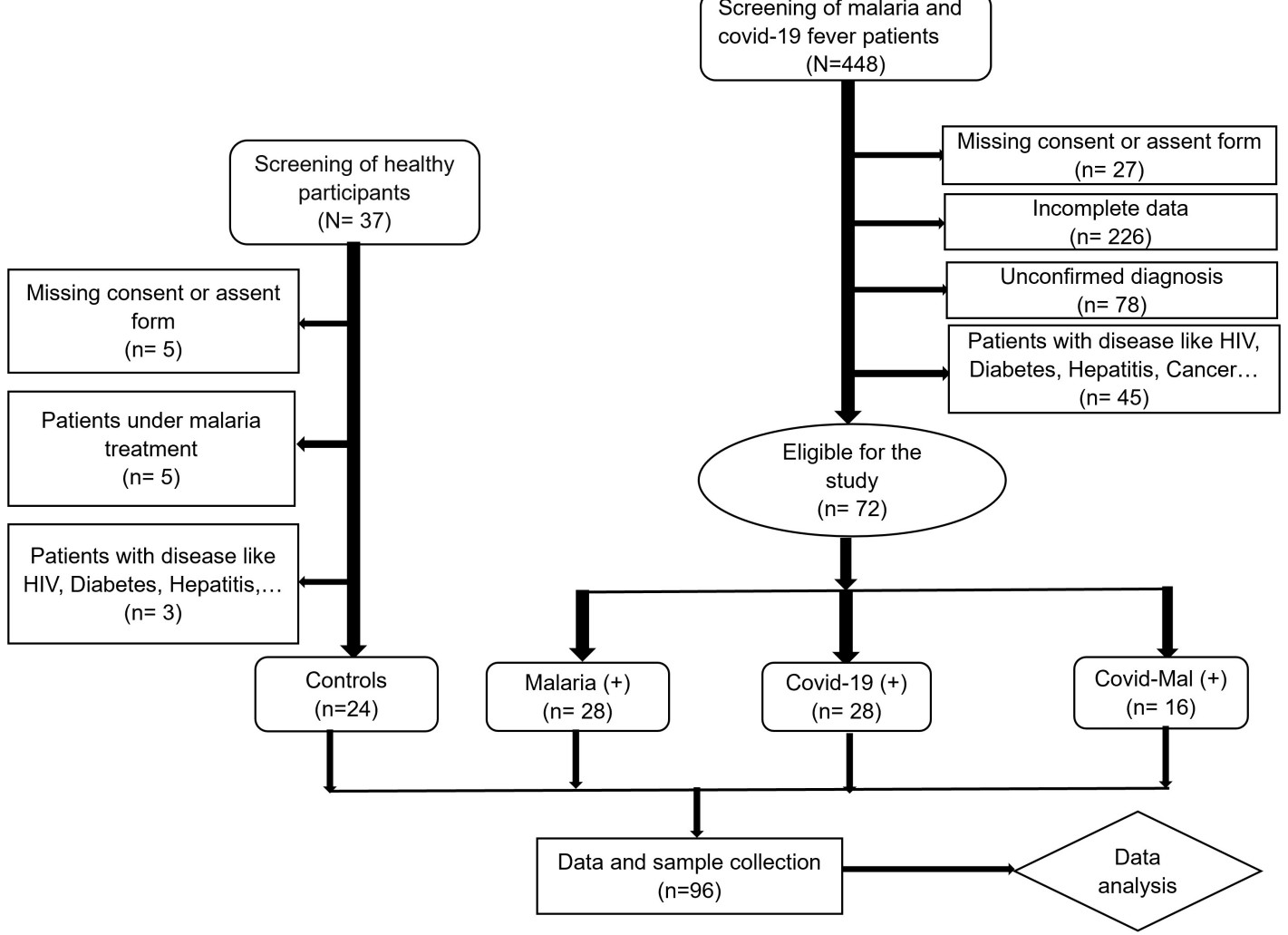

**Fig 1. Study flowchart for selection and clustering of participants.**

## Sample collection

Symptoms were self-reported by study participants to the clinician at the time of enrolment. Trained laboratory personnel collected approximately 5 mL of blood from each participant into both EDTA and plain tubes using sterile syringes. The samples were centrifuged at 3000 rpm for 5 minutes to separate serum and plasma. Nasopharyngeal swabs were also collected and placed in viral transport medium for COVID-19 diagnosis. The red blood cell pellets, nasopharyngeal swab samples, and aliquoted serum/plasma were transported to the National Public Health Laboratory and stored at −80°C for further analysis at the Molecular Diagnostic Research Laboratory (MDR-Lab) in Yaoundé.

## Diagnosis and confirmation of *Plasmodium falciparum*

*Plasmodium falciparum* antigen testing was performed using a rapid diagnostic test (RDT) based on immunochromatography. The SD Bioline Malaria Ag *P.f*/Pan kit (Standard Diagnostics, Inc.) which detects histidine-rich protein 2 (HRP-2), parasite lactate dehydrogenase, and aldolase, was used. The procedure involved applying two drops of blood and one drop of

buffer to the sample well, with results interpreted after ten minutes [22]. Parasitaemia was quantified by counting parasites per 200 white blood cells and adjusting the total white blood cell count [22]. Thick blood smear readings were performed in a double-blind manner, with a third reading conducted when discrepancies occurred. DNA extraction was carried out using the ZYMO Research Quick-DNA Miniprep Kit, followed by confirmatory diagnosis with a real-time RT-PCR test based on Co-Primers™ technology targeting the mitochondrial cytochrome C oxidase III (cox3) gene, selected for its higher copy number per parasite genome. RT-PCR was conducted on a Co-Dx Box™-advanced qPCR cycler (CO-DIAGNOSTICS, USA) employing Saragene malaria Real-Time PCR test kit (COSARA DIAGNOSTICS Pvt. Ltd, India).

### Diagnosis and confirmation of covid-19 infection

SARS-CoV-2 antigen testing was performed using immunochromatographic assays (Standard Q COVID-19 Ag and Standard Q COVID-19 IgG/IgM) from SD BIOSENSOR, Korea in 2020. Three drops of the sample were added to the test device, and results were interpreted after 15 minutes [23]. Confirmation of COVID-19 cases was based on the detection of viral RNA through Co-Primers™-based real-time RT-PCR assays. RNA was extracted using the QIAamp RNA Kit, and amplification was conducted on Co-Dx Box™-advanced qPCR cycler (CO-DIAGNOSTICS, USA) employing Co-Dx™ Logix Smart® SARS-CoV-2 (genes RdRp/E) test kit to detect the *RdRp* and *E* genes of SARS-CoV-2 [24].

### Measurement of liver and kidney function markers

Liver and kidney function markers were assessed in serum samples using spectrophotometric methods. Levels of SGOT, SGPT, bilirubin, urea, and creatinine were measured using reagent kits from Precise Max and Genuine Biosystem. Three analytical approaches were applied: the kinetic method, which monitored changes in absorbance over time to quantify creatinine [25], SGOT, and SGPT [26]; the endpoint method, which measured bilirubin concentration upon the completion of the reaction [27]; and the Berthelot method, which quantified urea by converting ammonia produced from urea hydrolysis into a coloured indophenol compound through reaction with phenol and hypochlorite ions under alkaline conditions [28].

### Measurement of inflammatory cytokine and circulating ACE2 and ANG2 levels

Inflammatory cytokine levels in plasma samples were quantified using antigen-capture ELISA kits from Melsin Medical Co., targeting seven cytokines analyzed in duplicate: IL-1β (Cat#: EKHU-0083), IL-2 (Cat#: EKHU-0144), IL-4 (Cat#: EKHU-0014), IL-6 (Cat#: EKHU-0140), IL-10 (Cat#: EKHU-0155), TNF-α (Cat#: EKHU-0110), and IFN-γ (Cat#: EKHU-1695). Additionally, ACE2 and ANG2 levels were measured in plasma samples using specific ELISA kits from RayBiotech (ACE2 Cat#: ELH-ACE2; ANG2 Cat#: ELH-ANG) [29,30]. Cytokine and vascular markers concentrations were calculated from optical density (OD) values using standard curves generated from known concentration standards.

### Data analysis

Statistical analyses were performed using SPSS version 26 and GraphPad Prism version 9.0.0. Prior to group comparisons, data distribution was assessed using the Shapiro–Wilk test to evaluate normality. For normally distributed data, group comparisons were conducted using one-way ANOVA, with data distributions visualized using boxplots for each patient group. When normality assumptions were not met, the Kruskal–Wallis test followed by appropriate post-hoc pairwise comparisons (e.g., Dunn's test) was performed as a non-parametric alternative. Associations between biomarker levels and clinical parameters were evaluated using correlation analyses. A p-value $< 0.05$ was considered statistically significant

## Results

### General characteristics of participants

The study included 96 participants recruited from four health facilities in Yaoundé, consisting of 72 symptomatic participants (mean age: 34.3 years) and 24 healthy controls. The most frequently reported symptoms were fever (63.54%),

headache (47.8%), and cough (42.7%). A significant difference in median parasite density was observed between the malaria (+) and malaria/COVID-19 co-infected groups (p < 0.001), suggesting that malaria severity may influence clinical outcomes and symptom presentation, particularly when co-occurring with COVID-19 (Table 1).

**Impact of malaria and COVID-19 on haemoglobin, platelets and leucocyte**

Malaria and COVID-19 negatively affect haematological parameters compared to healthy controls, particularly haemoglobin and platelet levels. Leukocyte counts were significantly higher in the malaria-only and COVID-19-only groups compared to controls (p = 0.04) (Table 2). Haematocrit levels were significantly lower in malaria participants relative to controls (p = 0.002). Platelet counts were markedly reduced in both malaria and COVID-19 patients compared to controls

**Table 1. Baseline demographics and clinical characteristics of the study population.**

| Characteristics | Control (n = 24) | Malaria (+) (n = 28) | Covid-19 (+) (n = 28) | Malaria-Covid-19 (n = 16) | p-value |
|---|---|---|---|---|---|
| Age in years (mean ±SD) | 30.45±6.7 | 25.4±16.2 | 37.25±12.83 | 40.25±12.25 | **0.7** |
| Male n (%) | 9 (45.8) | 11 (39) | 13 (46) | 10 (63) | |
| Female n (%) | 15 (54.2) | 17 (61) | 15 (54) | 6 (37) | |
| Temperature °C (mean ±SD) | 36.98±0.52 | 37.4±0.99 | 37.4±0.72 | 38.04±0.91 | 0.68 |
| Clinical parameters | | | | | |
| Fever n (%) | / | 16 (57) | 22 (78.6) | 11 (68.8) | 0.75 |
| Cough n (%) | / | 10 (35.7) | 20 (71.4) | 11 (68.8) | **0.012*** |
| Headache n (%) | / | 17 (60.7) | 18 (64.2) | 8 (50) | 0.50 |
| Fatigue n (%) | / | 14 (50) | 12 (42.8) | 5 (31.25) | 0.23 |
| Rhinorrhoea n (%) | / | 0 (0) | 11 (39.3) | 3 (18.75) | **<0.0001*** |
| Sore throat n (%) | / | 0 (0) | 7 (25) | 2 (12.5) | **0.003*** |
| Chest pain n (%) | / | 6 (23) | 8 (28.5) | 3 (18.75) | 0.83 |
| Myalgia n (%) | / | 16 (57) | 11 (39.3) | 2 (12.5) | **0.02*** |
| Arthralgia n (%) | / | 9 (32) | 12 (42.8) | 5 (31.25) | 0.63 |
| Loos of smell n (%) | / | 0 (0) | 8 (28.6) | 1 (6.25) | **0.0016*** |
| Loss of flavour n (%) | / | 0 (0) | 3 (10.7) | 1 (6.25) | 0.16 |
| Abdominal pain n (%) | / | 11 (39.3) | 5 (17.8) | 1 (6.25) | **0.04*** |
| Confusion n (%) | / | 0 (0) | 1 (3.6) | 1 (6.25) | 0.36 |
| Convulsion n (%) | / | 2 (7.1) | 2 (7.1) | 0 (0) | 0.59 |
| Vomiting n (%) | / | 9 (32.1) | 1 (3.6) | 0 (0) | **0.007*** |
| Diarrhoea n (%) | / | 3 (10.7) | 0 (0) | 0 (0) | 0.14 |
| Skin rash n (%) | / | 0 (0) | 2 (7.1) | 0 (0) | 0.17 |
| Skin ulcers n (%) | / | 0 (0) | 1 (3.6) | 0 (0) | 0.42 |
| Heart rate bpm (mean ±SD) | 88.45±16.38 | 81.89±12.42 | 87.34±10.96 | 82.75±9.66 | 0.14 |
| Respiratory rate bm (mean ±SD) | 18.87±1.32 | 18.44±1.73 | 16.83±2.8 | 16.43±1.75 | **<0.0001*** |
| Systolic blood pressure BP/mmHg (mean ±SD) | 117.83±10.89 | 116.89±12.28 | 122.46±19.72 | 117.5±12.07 | 0.47 |
| Diastolic blood pressure BP/mmHg (mean ±SD) | 76.95±8.78 | 78.58±9.98 | 79.96±13.81 | 76.18±10.21 | 0.66 |
| SpO2 saturation % (mean ±SD) | 98.04±1.33 | 97.05±1.75 | 97 ± 167 | 96.37±1.82 | **0.01*** |
| Median parasite density/μl of blood | / | 508.62 (40-1550) | | 1041.73 (75-3000) | **<0.001*** |

Data represented as count (percentage, %), mean (standard deviation) and range (min - max) where the asterisk (*) corresponds to statistical significance at p < 0.05

 

**Table 2. Hematological parameters.**

| Blood parameter | Control (n = 24) | Malaria (+) (n = 28) | Covid-19 (+) (n = 28) | Malaria-Covid-19 (n = 16) | p-value |
|---|---|---|---|---|---|
| Leukocyte (mean ± SD) x10⁹/L | 4.78 ± 1.2 | 5.86 ± 2.93 | 7.2 ± 4.32 | 5.43 ± 3.38 | 0.04* |
| Hematocrit, (Sd) % | 41.77 ± 6 | 35.26 ± 5.12 | 42.16 ± 5.52 | 40.83 ± 7.76 | 0.002* |
| Neutrophils (SD) X10⁹/L | 2.22 ± 1.16 | 2.7 ± 1.08 | 2.43 ± 1.37 | 2.8 ± 0.89 | 0.44 |
| Lymphocytes (SD) X10⁹/L | 2.3 ± 1.1 | 1.54 ± 1.47 | 2.27 ± 1.63 | 2.01 ± 1.16 | 0.11 |
| Platelets (SD) X10⁹/L | 280.81 ± 91.5 | 197.61 ± 100.25 | 188.8 ± 70.61 | 185.87 ± 65.6 | 0.0005* |
| Hemoglobin (SD), g/dl | 14.27 ± 1.86 | 11.94 ± 1.82 | 13.66 ± 1.93 | 12.01 ± 2.84 | 0.0006* |

Legend: Data represented as mean (standard deviation) where the asterisk (*) corresponds to statistical significance compared with control. *p < 0.05, **p < 0.01, ***p < 0.001.

(p = 0.0005). Haemoglobin levels were also significantly decreased in the malaria (+) and malaria/COVID-19 co-infected groups (p = 0.0006). No significant differences were observed in neutrophil and lymphocyte counts between the groups (p = 0.44 and p = 0.11, respectively).

### Increased AST, ALT, urea, creatinine, erythropoietin, and bilirubin are associated with co-infection

Biochemical parameters related to liver and kidney function were evaluated in all study participants (Fig 2). Levels of AST, ALT, urea, creatinine, erythropoietin, and bilirubin were significantly higher in the co-infected group compared to the mono-infected and healthy control groups (p < 0.05). No significant differences in D-dimer levels were observed between the groups.

### Increased IL-1β and IFN-γ levels linked to *Plasmodium falciparum* and SARS-CoV-2 coinfection

Plasma IFN-γ, TNF-α, IL-6, IL-10, IL-4, IL-2, and IL-1β levels were compared across the study groups (Fig 3). Plasma IFN-γ and IL-1β levels were significantly higher in malaria/COVID-19 co-infected group compared with both healthy controls and mono-infected groups (p < 0.001). Additionally, IL-6, IL-4, and IL-10 levels were elevated in malaria positive participants relative to controls (p < 0.01). While IL-2 and TNF-α levels were higher in COVID-19 positive participants compared to controls (p < 0.0001).

### ACE2 and ANG2 are significantly higher in both the COVID-19 mono-infection and co-infection group

Circulating ANG2 levels in the COVID-19, malaria, and co-infected groups were higher than those in the healthy control group (p < 0.01). In contrast, ACE2 levels were elevated in the COVID-19 group compared to the malaria, co-infected, and healthy control groups (p < 0.01). (Fig 4).

### Cytokine and biomarker correlations with parasitaemia and Ct-values

In this study, we examined the association between malaria parasitaemia and cytokine levels in malaria and co-infected groups, as well as the relationship between SARS-CoV-2 Ct-values and cytokines in the COVID-19 group (Table 3). Among the malaria group, ACE2 plasma levels showed a strong negative correlation with parasitaemia (r = −0.60, p = 0.0002). IL-4 levels were also negatively correlated with parasitaemia (r = −0.391, p = 0.04), whereas erythropoietin and D-dimer levels were positively correlated (erythropoietin: r = 0.426, p = 0.02; D-dimer: r = 0.417, p = 0.03). In the COVID-19 group, IL-4 levels were positively correlated with Ct-values (r = 0.532, p = 0.003). In the malaria/COVID-19 co-infected group, ANG2, IL-4, and erythropoietin levels each showed positive correlations with parasitaemia (ANG2: r = 0.558, p = 0.02; IL-4: r = 0.536, p = 0.03; erythropoietin: r = 0.585, p = 0.02).

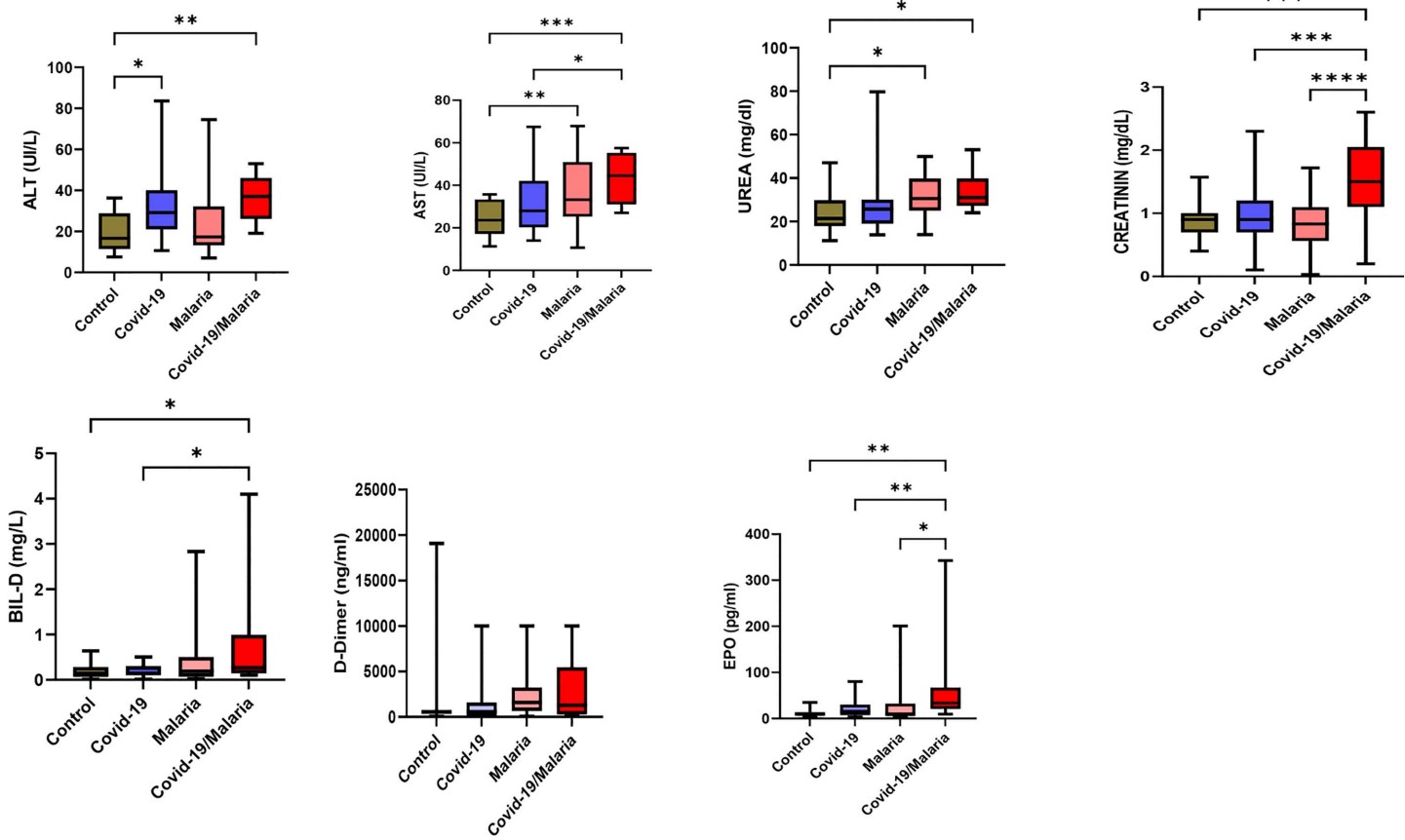

**Fig 2. Biochemical parameters related to liver and kidney function among study groups.** *p < 0.05, **p < 0.01, ***p < 0.001.

### Receiver Operating Characteristic analysis of ACE2 and ANG2 for predicting disease severity

Receiver operating characteristic analysis was used to assess the diagnostic performance of ANG2 and ACE2 in predicting disease severity. The area under the curve (AUC), sensitivity, specificity, p-values, and optimal cut-off values for the different study groups are presented in Table 4.

The areas under the curve (AUC) for ACE2 in the malaria and COVID-19 groups were 0.77 (95% CI: 0.59–0.94) and 0.85 (95% CI: 0.67–1.00), respectively (Table 4, Fig 5). ROC analysis identified ACE2 cut-off values of 3.29 pg/mL for malaria and 4.67 pg/mL for COVID-19. The sensitivity and specificity of ACE2 were 81% and 71.4% for malaria, and 66.7% and 92% for COVID-19, with statistically significant results (p < 0.05). Overall, the AUCs indicate good predictive performance for disease severity in both COVID-19 and malaria (Table 4).

### Discussion

This study aimed to better understand how coinfection alters cytokine profiles, vascular and biochemical markers, potentially influencing the severity of both diseases. Both malaria and COVID-19 significantly alter blood counts compared to healthy controls. We found higher overall white-cell (leukocyte) counts in malaria-only and COVID-19-only patients (p = 0.04), which aligns with malaria's known leukocyte changes and COVID-19's potential for leucocytosis [31,32]. Haematocrit was significantly lower in malaria patients than controls (p = 0.002), reflecting malaria's effect on red cells

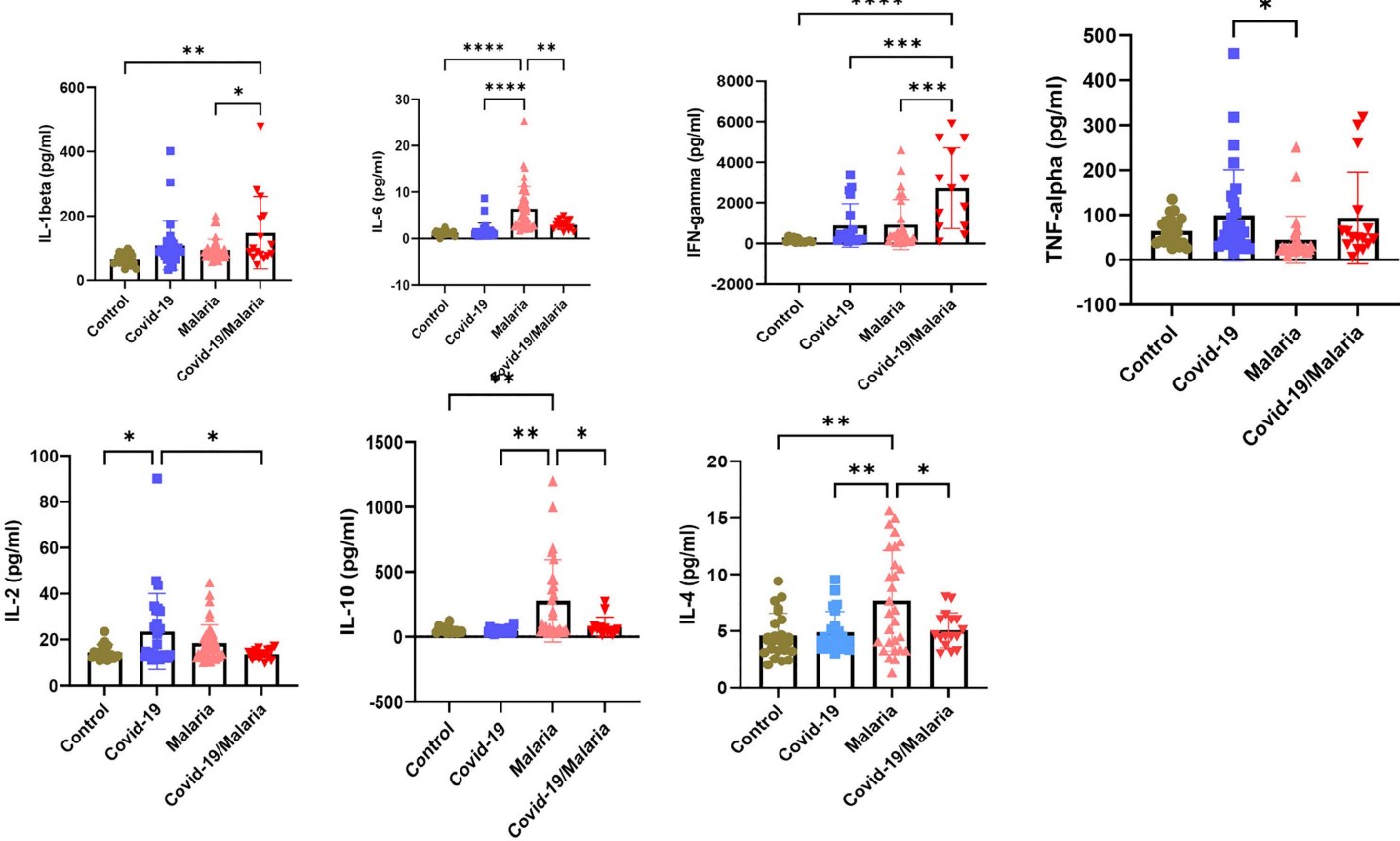

**Fig 3. Serum cytokine levels (IFN-γ, TNF-α, IL-6, IL-10, IL-4, IL-2, and IL-1β) among study groups. *p<0.05, **p<0.01, ***p<0.001.**

and anaemia [33]. Platelet counts were notably lower in both malaria and COVID-19 groups compared with controls (p=0.0005); this matches known thrombocytopenia in both diseases [34]. Haemoglobin levels also dropped significantly in the malaria and malaria-COVID-19 groups (p=0.0006), again mirroring the anaemia seen in malaria. No meaningful differences were observed in neutrophil or lymphocyte counts (p=0.44 and p=0.11), suggesting these specific white-cell subsets may not be greatly altered in our cohort despite the overall leukocyte changes.

In our study, individuals co-infected with malaria and COVID-19 showed significantly higher levels of liver enzymes (AST and ALT), elevated kidney function markers (urea and creatinine), increased bilirubin, and higher erythropoietin levels compared to those who had a single infection or were healthy controls (p<0.05). These findings suggest that co-infection places additional stress on both liver and kidney function. Previous research supports this observation, as both malaria–virus co-infections and COVID-19 alone have been shown to impair hepatic and renal function [35]. Interestingly, D-dimer levels did not differ among the groups, indicating that while organ stress was evident, coagulation dysfunction was not a distinguishing feature in this cohort.

Cytokine profiling revealed distinct immune signatures across infection groups, reflecting the combined pathogenic pressure of malaria and COVID-19. Significantly elevated IFN-γ and IL-1β in co-infected individuals indicate heightened Th1-mediated responses, consistent with reports that both *Plasmodium* species and SARS-CoV-2 trigger strong pro-inflammatory pathways contributing to disease severity [36,37]. Increased IL-6, IL-4, and IL-10 among malaria-positive participants aligned with previous studies attributing these cytokines to immune regulation during blood-stage malaria, where IL-10 and IL-4 counterbalance excessive inflammation [38]. Conversely, elevated IL-2 and TNF-α in COVID-19

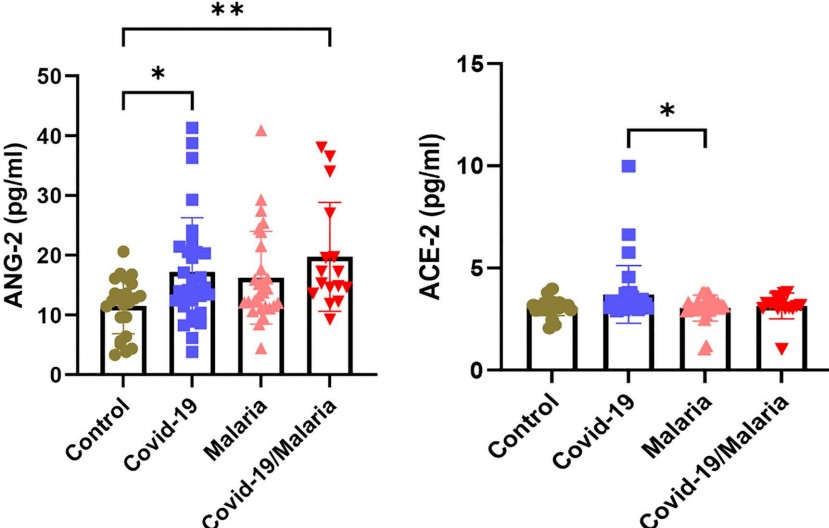

**Fig 4. Circulating ANG2 and ACE2 Levels among Study Groups.**

**Table 3. Spearman correlation analysis between parasitemia/Ct value and biomarkers.**

| | Malaria parasitemia (28) | | Malaria/covid-19 Parasitemia (28) | | Covid-19 Ct-value (24) | |
|---|---|---|---|---|---|---|
| | ρ | p-value | ρ | p-value | ρ | p-value |
| IL-2 | 0.181 | 0.3 | -0.338 | 0.2 | 0.209 | 0.2 |
| IL-6 | 0.205 | 0.3 | -0.242 | 0.3 | -0.0005 | 0.9 |
| IL-1β | -0.049 | 0.8 | 0.166 | 0.5 | 0.086 | 0.6 |
| INF-γ | 0.122 | 0.5 | 0.119 | 0.6 | -0.066 | 0.7 |
| ACE2 | -0.654 | 0.0002* | -0.034 | 0.9 | 0.078 | 0.6 |
| ANG2 | 0.196 | 0.3 | 0.588 | 0.02* | -0.284 | 0.1 |
| IL-10 | -006 | 0.9 | 0.384 | 0.1 | -0.1 | 0.6 |
| EPO | 0.426 | 0.02* | 0.536 | 0.03* | -0.291 | 0.1 |
| IL-4 | -0.391 | 0.04* | 0.585 | 0.02* | 0.532 | 0.003* |
| TNF-α | 0.017 | 0.9 | -0.285 | 0.3 | -0.141 | 0.4 |
| d-dimer | 0.417 | 0.03* | -0.042 | 0.8 | -0.057 | 0.7 |

*Legend: Asterisk (\*) represent a statistically significant associations with p < 0.05*

mono-infected individuals supported evidence linking these cytokines to T-cell activation and systemic inflammation during SARS-CoV-2 infection [39]. The overlapping yet distinct cytokine elevations in co-infection underscore a potential synergistic immune dysregulation that may predispose individuals to worse clinical outcomes.

Circulating Angiopoietin-2 (Ang-2) levels were elevated across the COVID-19, malaria and co-infected groups compared with healthy controls. This aligns closely with the literature demonstrating that elevated Ang-2 has been linked to increased ICU admission, acute kidney injury and mortality in COVID-19 patients [40]. The higher Ang-2 in malaria and co-infection suggests similar endothelial stress may occur in malaria alone and might be compounded in co-infection. Meanwhile, our finding that Angiotensin-converting enzyme 2 (ACE2) levels were specifically higher in the COVID-19 only group (compared to malaria, co-infected and controls). is consistent with studies showing altered

PLOS Global Public Health

**Table 4. ROC analysis of ANG2 and ACE2 biomarkers for predicting disease severity in covid-19 and malaria.**

| | | Optical cut-of value (pg/ml) | AUC (%) | Sensitivity (%) | Specificity (%) | p-value |
|---|---|---|---|---|---|---|
| Malaria (+) | ANG2 | <19.67 | 65.3 | 76.2 | 71.4 | 0.20 |
| | ACE2 | <03.29 | 77.2 | 81.0 | 71.4 | 0.03* |
| Covid-19 (+) | ANG2 | <25.66 | 76.0 | 66.7 | 96.0 | 0.10 |
| | ACE2 | <04.67 | 85.3 | 66.7 | 92.0 | 0.04* |

Legend: * Statistical significance, AUC – Area under the curve.

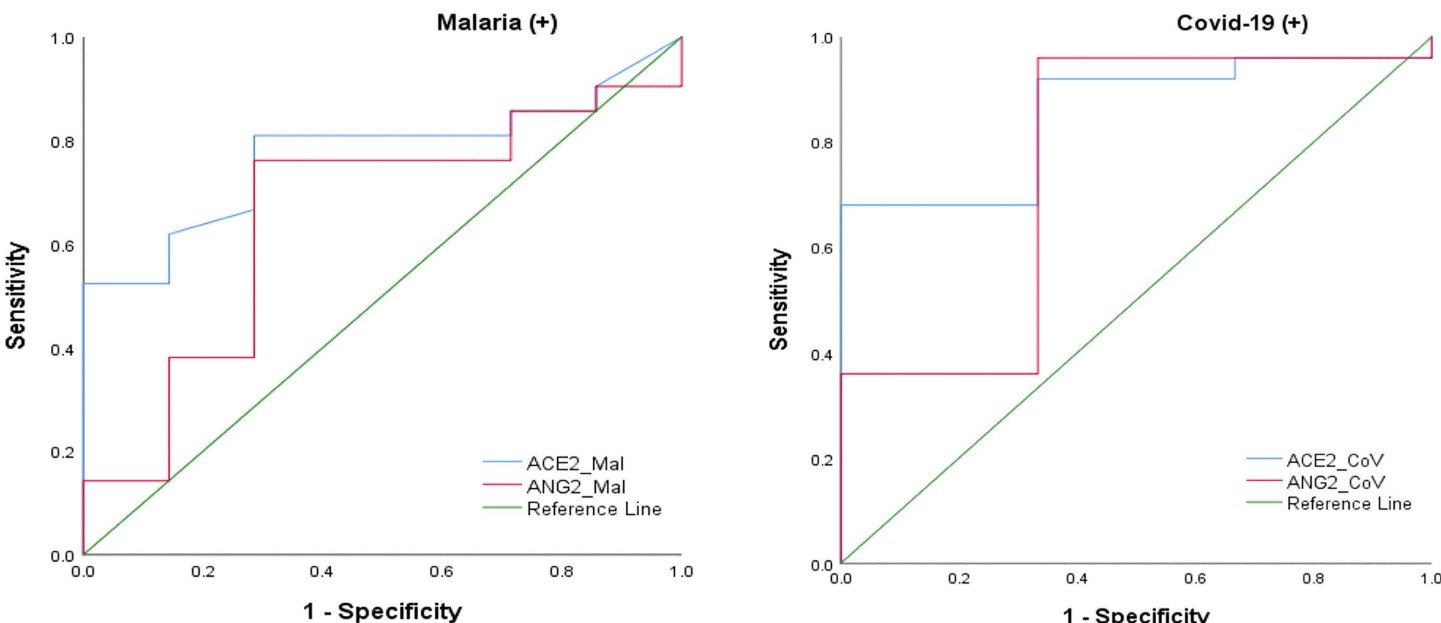

**Fig 5. Comparison of ROC curves for predicting disease severity.**

ACE2 expression in COVID-19 and potential interactions with the renin-angiotensin system in malaria endemic settings [41].

In the malaria group, the strong negative correlation between plasma ACE2 levels and parasitaemia (r= −0.60, p=0.0002), aligns with prior work showing that reduced ACE2 expression (or higher gene polymorphism activity) is linked to increased Ang-II and may influence malaria susceptibility or severity via the renin-angiotensin system [42]. The negative correlation between IL-4 with parasitaemia (r= −0.391, p=0.04) is consistent with a study showing that higher IL-4 was associated with lower parasite load in Plasmodium vivax infections [43]. Meanwhile, the positive correlations between parasitaemia and erythropoietin (r=0.426, p=0.02) as well as D-dimer (r=0.417, p=0.03) reflect reports that higher parasite density is associated with increased erythropoietic drive and coagulation activation in malaria [44]. In the COVID-19 group, the positive correlation between IL-4 and Ct-value (r=0.532, p=0.003) suggests that higher IL-4 is associated with lower viral load (higher Ct), a pattern that complements findings linking elevated IL-4 to less severe SARS-CoV-2 infection although direct Ct correlations are sparse. Finally, in the co-infected group the positive correlations for ANG2 (r=0.558, p=0.02), IL-4 (r=0.536, p=0.03), and erythropoietin (r=0.585, p=0.02) with parasitaemia point toward synergistic vascular and hematologic responses when malaria and COVID-19 coincide. These comparative findings suggest that parasite

load and viral load may each modulate distinct immune and endothelial biomarkers, emphasizing the need for joint infection studies.

ACE2 showed good discrimination for severe disease in both malaria (AUC = 0.77; 95% CI 0.59–0.94; cut-off 3.29 pg/mL; sensitivity 81%; specificity 71.4%) and COVID-19 (AUC = 0.85; 95% CI 0.67–1.00; cut-off 4.67 pg/mL; sensitivity 66.7%; specificity 92%). These findings align with previous COVID-19 studies, where elevated ACE2 levels were associated with worse outcomes and reported AUCs around 0.70. However, there have been few prior reports of ACE2 being used in this manner in malaria, so our results add new evidence to that field [45].

## Conclusion

This study demonstrated that malaria and COVID-19, particularly when co-occurring, disrupt key haematological, biochemical, and immune pathways contributing to disease severity. Significant reductions in haemoglobin, haematocrit, and platelet levels highlight anaemia and thrombocytopenia as common consequences of infection, while co-infection further exacerbates liver and kidney dysfunction, indicating greater systemic stress. Immune profiling reveals that malaria–COVID-19 co-infection triggers heightened inflammatory responses, particularly through elevated IFN-γ and IL-1β, suggesting synergistic immune dysregulation. Elevated IL-6, IL-4, and IL-10 in malaria cases, alongside increased IL-2 and TNF-α in COVID-19, highlight pathogen-specific immune regulation. Vascular injury markers were also strongly affected: ANG2 was elevated across all infected groups, reflecting endothelial activation, whereas ACE2 was specifically increased in COVID-19, consistent with its role in viral entry and renin–angiotensin system imbalance. Correlation patterns further indicate that parasitaemia and viral load influence distinct biomarker pathways linked to inflammation, erythropoiesis, and endothelial stress. Finally, ROC analysis identified ACE2 and ANG2 as promising severity predictors, with good diagnostic performance in both malaria and COVID-19. Collectively, these findings emphasize the clinical value of monitoring vascular and immune biomarkers to enhance early risk stratification, particularly in co-infected patients.

## Supporting information

**S1 Data. S1 Data presents a dataset designed for comparative analysis of clinical presentations, organ dysfunction, and immune responses across three distinct infectious disease states [individuals with COVID-19, MALARIA, and those with both malaria and covid-19 (MALCOV)] as well as a healthy Control group.** Each record includes patient demographics (Lab code, Sex, Age), vital signs (Temperature, Heart Rate, Blood Pressure, Oxygen Saturation), the presence or absence of 20 common symptoms (coded as 1/0, e.g., Fever, Cough, Loss of smell), standard blood biochemical markers (e.g., liver enzymes ALAT/ASAT, kidney function markers, D-Dimer), and a panel of nine immune and inflammatory cytokines (e.g., IL-2, IL-6, IL-1beta, INF-gamma, ACE-2, ANG-2, IL-10, IL-4 and TNF-alpha).
(XLSX)

## Author contributions

**Conceptualization:** Eric Berenger Tchoupe, Mary Ngongang Kameni, Anthony Afum-Adjei Awuah, Jonh Humphrey Amuasi, Palmer Netongo.

**Data curation:** Eric Berenger Tchoupe, Mary Ngongang Kameni, Severin Donald Kamdem.

**Formal analysis:** Eric Berenger Tchoupe, Mary Ngongang Kameni, MacDonald Bin Eric, Severin Donald Kamdem.

**Funding acquisition:** Jonh Humphrey Amuasi, Palmer Netongo.

**Investigation:** Eric Berenger Tchoupe, Mary Ngongang Kameni, Arnaud Tepa, Fuh Roger Neba.

**Methodology:** Eric Berenger Tchoupe, Mary Ngongang Kameni, MacDonald Bin Eric, Jean Bosco Taya, Severin Donald Kamdem, Leonard Numfor Nkah, Vicky Ama Moor, Arnaud Tepa, Fuh Roger Neba, Palmer Netongo.

**Project administration:** Vicky Ama Moor, Anthony Afum-Adjei Awuah, Jonh Humphrey Amuasi, Palmer Netongo.

**Resources:** Anthony Afum-Adjei Awuah, Jonh Humphrey Amuasi, Palmer Netongo.

**Supervision:** Vicky Ama Moor, Palmer Netongo.

**Validation:** Eric Berenger Tchoupe, Mary Ngongang Kameni, MacDonald Bin Eric, Jean Bosco Taya, Severin Donald Kamdem, Leonard Numfor Nkah, Vicky Ama Moor, Arnaud Tepa, Fuh Roger Neba, Anthony Afum-Adjei Awuah, Jonh Humphrey Amuasi, Palmer Netongo.

**Writing – review & editing:** Eric Berenger Tchoupe, Mary Ngongang Kameni, MacDonald Bin Eric, Jean Bosco Taya, Severin Donald Kamdem, Leonard Numfor Nkah, Vicky Ama Moor, Arnaud Tepa, Palmer Netongo.

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
