## [Decision Letter · Decision Letter 0]

16 Oct 2025

PGPH-D-25-01398

Investigating the potential of clinical variables and ace2/tmprss2 genetic variants as a biomarkers for COVID-19 and malaria co-infection in Cameroon

Dear Dr. Kamoua,

Thank you for submitting your manuscript to PLOS Global Public Health. After careful consideration, we feel that it has merit but does not fully meet PLOS Global Public Health’s publication criteria as it currently stands. Therefore, we invite you to submit a revised version of the manuscript that addresses the points raised during the review process.

We look forward to receiving your revised manuscript.

Kind regards,

Abhinav Sinha, M.D.

Academic Editor

Journal Requirements:

1. Your current Financial Disclosure states, “This study was funded by the African coaLition for Epidemic Research, Response and Training (ALERRT), part of the EDCTP2 Programme supported by the European Union”. However, your funding information on the submission form indicates that you received funding from “FP7 Ideas: European Research Council”. Please indicate by return email the full and correct funding information for your study and confirm the order in which funding contributions should appear. Please be sure to indicate whether the funders played any role in the study design, data collection and analysis, decision to publish, or preparation of the manuscript.

3. Some material included in your submission may be copyrighted. According to PLOS’s copyright policy, authors who use figures or other material (e.g., graphics, clipart, maps) from another author or copyright holder must demonstrate or obtain permission to publish this material under the Creative Commons Attribution 4.0 International (CC BY 4.0) License used by PLOS journals. Please closely review the details of PLOS’s copyright requirements here: PLOS Licenses and Copyright. If you need to request permissions from a copyright holder, you may use PLOS's Copyright Content Permission form.

Potential Copyright Issues:

Figure 1: please (a) provide a direct link to the base layer of the map (i.e., the country or region border shape) and ensure this is also included in the figure legend; and (b) provide a link to the terms of use / license information for the base layer image or shapefile. We cannot publish proprietary or copyrighted maps (e.g. Google Maps, Mapquest) and the terms of use for your map base layer must be compatible with our CC-BY 4.0 license.

4. We do not publish any copyright or trademark symbols that usually accompany proprietary names, eg (R), (C), or TM  (e.g. next to drug or reagent names). Please remove all instances of trademark/copyright symbols throughout the text, including ®, ™ on page 5, 6.

Reviewers' comments:

Reviewer's Responses to Questions

**Comments to the Author**

1. Does this manuscript meet PLOS Global Public Health’s publication criteria?

Reviewer #1: No

Reviewer #2: Partly

2. Has the statistical analysis been performed appropriately and rigorously?

Reviewer #1: N/A

Reviewer #2: I don't know

3. Have the authors made all data underlying the findings in their manuscript fully available (please refer to the Data Availability Statement at the start of the manuscript PDF file)?

Reviewer #1: Yes

Reviewer #2: Yes

4. Is the manuscript presented in an intelligible fashion and written in standard English?

Reviewer #1: No

Reviewer #2: No

Reviewer #1: The research topic is interesting but it has major concerns. The research work reported here does not fully align with the title of the paper. In general the introduction, results and discussion are not in sync with the title, unable to give a clear understanding to the reader about the biomarkers or genetic variants for both the diseases. In light of these comments I do not recommend this paper.

Other comments:

1. As per the title of the paper “Investigating the potential of clinical variables and ace2/tmprss2 genetic variants as a biomarkers for COVId-19 and malaria co-infection Cameroon,” does not explain the genetic variables in a clear manner in the introduction nor in the discussion.

2. Short title has error ‘alta’ which needs to be corrected.

3. The scientific names are not italicized eg, Plasmodium and several places Covid-19 is written in different way thought out the manuscript.

Introduction-

4. In Introduction the background is vague. No mentioned of any previous SNP based study is there.

5. Which SNPs are to be studied do not find mention in the introduction.

6. Line 91 – ‘Trained immunity’. Please rephrase the sentence.

7. Hypothesis is not clear. Pl revise.

8. Aims need to be modified.

M & M-

9. The subheadings are very long. They need to be crisper.

10. Line 126 - - ‘as if 2015’ this is an old data date, to be replaced with current.

11. Full stops to be added after completion of sentences.

12. Line 154 – ‘malaria tests were ordered’ – It is not clear which test were undertaken.

13. Line 157 – ‘aliquoted samples’ – not clear with aliquoted samples.

14. Line 168 ‘RT-PCR for cox-3 gene’ done Any particular reason why this test was used. Explain it.

15. No clear mention of which species was studied here.

16. Title mentions malaria patients so it is expected that Pf and non- Pf species should all be included here, but this does not seem so. Pl explain this clearly.

17. Line 176 no reference mentioned.

Results -

18. Table -1 ‘Malaria mono’ specify clearly.

19. Headings are too long, needs to be shortened. Heading itself is stating the results.

20. Line 319 – homo, heterozygous and homozygous mutant not explained properly.

Discussion –

21. Too long. Needs to be fully revised.

22. Lines 377-380. Contradicting and derived statement needs to be revisited.

23. Lines 386-387. The link of malaria, AST ALT increased levels with Schistosomiasis is not clearly understandable.

Reviewer #2: 1. English writing is poor, there are grammatical mistake throughout the text. Authors should improve the English writing. This manuscript is required the major revision.

2. The sentence “ Malaria is a tropical disease caused by parasites transmitted through bites from Plasmodium-infected mosquitoes resulting in approximately 249 million cases and 608,000 deaths annually, particularly in sub-Saharan Africa” needs to be reframed. It is not telling about the mosquito species. Please mention WHO report and year regarding malaria global burden and deaths.

3. Please cite , any meta -analysis or original research work on the trained immunity induced by previous malaria exposure, which protects against covid-19 severity.

4. Covid severity is caused by due cytokine storm of pro-inflammatory cytokines. How can pro-inflammatory cytokines induce during malaria protect against covid severity. Line 96-98

5. Authors should use the reference, alternate genotypes in place of wild and mutant. Line 318-19

6. Authors should add the study design and prevalence of malaria and covid-19 in the Cameroon.

**Do you want your identity to be public for this peer review?** For information about this choice, including consent withdrawal, please see our Privacy Policy

Reviewer #1: No

Reviewer #2: No

---

## [Decision Letter · Decision Letter 1]

16 Dec 2025

PGPH-D-25-01398R1

Elevated ANG2/ACE2 and immune responses associated with Plasmodium falciparum and SARS-CoV-2 coinfection in Cameroon

Dear Dr. Netongo,

Thank you for submitting your manuscript to PLOS Global Public Health. After careful consideration, we feel that it has merit but does not fully meet PLOS Global Public Health’s publication criteria as it currently stands. Therefore, we invite you to submit a revised version of the manuscript that addresses the points raised during the review process.

We look forward to receiving your revised manuscript.

Kind regards,

Abhinav Sinha, M.D.

Academic Editor

Journal Requirements:

1. Please amend your detailed online Financial Disclosure statement. This is published with the article. It must therefore be completed in full sentences and contain the exact wording you wish to be published.

a) State the initials, alongside each funding source, of each author to receive each grant. For example: “This work was supported by the National Institutes of Health (####### to AM; ###### to CJ) and the National Science Foundation (###### to AM).”

For more information, please go to our submission guidelines:

https://journals.plos.org/globalpublichealth/s/submission-guidelines#loc-financial-disclosure-statement

2. Please ensure that the funders and grant numbers match between the Financial Disclosure field and the Funding Information tab in your submission form. Note that the funders must be provided in the same order in both places as well.

3. Please update your online Competing Interests statement. If you have no competing interests to declare, please state: “The authors have declared that no competing interests exist.”

4. We note that your Data Availability Statement is currently as follows: “all data for this manuscript are available”

Please confirm at this time whether or not your submission contains all raw data required to replicate the results of your study. Authors must share the “minimal data set” for their submission. PLOS defines the minimal data set to consist of the data required to replicate all study findings reported in the article, as well as related metadata and methods (https://journals.plos.org/globalpublichealth/s/data-availability#loc-minimal-data-set-definition).

If your submission does not contain these data, please either upload them as Supporting Information files or deposit them to a stable, public repository and provide us with the relevant URLs, DOIs, or accession numbers. For a list of recommended repositories, please see https://journals.plos.org/globalpublichealth/s/recommended-repositories.

5. Please ensure that you refer to Figure 1 in your text as, if accepted, production will need this reference to link the reader to the figure.

Additional Editor Comments (if provided):

The authors have revised the manuscript as per the reviewer's suggestions. However, one of the reviewers has suggested to add a couple of references. I would leave the decision on the authors whether they feel that adding the references would improve the manuscript.

Reviewers' comments:

Reviewer's Responses to Questions

**Comments to the Author**

Reviewer #2: All comments have been addressed

publication criteria?

Reviewer #2: Yes

3. Has the statistical analysis been performed appropriately and rigorously?

Reviewer #2: I don't know

4. Have the authors made all data underlying the findings in their manuscript fully available (please refer to the Data Availability Statement at the start of the manuscript PDF file)?

Reviewer #2: Yes

5. Is the manuscript presented in an intelligible fashion and written in standard English?

Reviewer #2: Yes

Reviewer #2: The authors should work on English writing.

**Do you want your identity to be public for this peer review?** For information about this choice, including consent withdrawal, please see our Privacy Policy

Reviewer #2: No

---

## [Editor Report · Decision Letter 2]

19 Jan 2026

Elevated ANG2/ACE2 and immune responses associated with Plasmodium falciparum and SARS-CoV-2 coinfection in Cameroon

PGPH-D-25-01398R2

Dear Dr Netongo,

We are pleased to inform you that your manuscript 'Elevated ANG2/ACE2 and immune responses associated with Plasmodium falciparum and SARS-CoV-2 coinfection in Cameroon' has been provisionally accepted for publication in PLOS Global Public Health.

Best regards,

Abhinav Sinha, M.D.

Academic Editor